# NeMo-Aligner: Scalable Toolkit for Efficient Model Alignment

**Gerald Shen, Zhilin Wang, Olivier Delalleau, Jiaqi Zeng, Yi Dong,**
**Daniel Egert, Shengyang Sun, Jimmy Zhang, Sahil Jain, Ali Taghibakhshi**
**Markel Sanz Ausin, Ashwath Aithal, Oleksii Kuchaiev**
NVIDIA
{geshen, zhilinw}@nvidia.com

## Abstract

Aligning Large Language Models (LLMs) with human values and preferences is essential for making them helpful and safe. However, building efficient tools to perform alignment can be challenging, especially for the largest and most competent LLMs which often contain tens or hundreds of billions of parameters. We create NeMo-Aligner, a toolkit for model alignment that can efficiently scale to a thousand GPUs for training the largest open-source LLMs such as Nemotron 4 340B and Llama 3.1 405B. NeMo-Aligner comes with highly optimized and scalable implementations for major paradigms of model alignment such as: Reinforcement Learning from Human Feedback (RLHF), Direct Preference Optimization (DPO), SteerLM, and Self-Play Fine-Tuning (SPIN). Additionally, our toolkit supports running most of the alignment techniques in a Parameter Efficient Fine-Tuning (PEFT) setting. NeMo-Aligner is designed for extensibility, allowing support for other alignment techniques with minimal effort. It is open-sourced with Apache 2.0 License and we invite community contributions at https://github.com/NVIDIA/NeMo-Aligner.

## 1 Introduction

Pre-training large language models on tremendous amounts of unlabelled text has showcased promising capabilities (Brown et al., 2020; Zhang et al., 2022). While such unsupervised pre-trained models have achieved impressive results, subsequently aligning models to follow user instructions is a critical step to tap the capabilities of LLMs for practical use cases (Sanh et al., 2022; Wei et al., 2022). Attempts based on Supervised Finetuning (Conover et al., 2023; Köpf et al., 2023; Taori et al., 2023) proved less effective compared to techniques that also made use of feedback to tune models towards responses that are more helpful and away from responses that are less so (Bai et al., 2022a; Ouyang et al., 2022; Touvron et al., 2023; Dong et al., 2023).

Despite the benefits of training models using feedback, these pipelines are notoriously challenging to get right (Lambert & Calandra, 2023; Zheng et al., 2023b), deterring widespread, productive adoption outside of select well-resourced organizations. For example, the popular Proximal Policy Optimization (PPO) variant of Reinforcement Learning from Human Feedback (RLHF) approach (Ouyang et al., 2022) requires running a complicated pipeline with four large language models interacting in a complex manner during training. Such alignment algorithms introduce new system challenges for efficient training that require re-thinking various aspects of the software stack including model scalability, coordination among models, and text generation within the training loop.

There are existing open source tools for model alignment, most notably HuggingFace TRL (von Werra et al., 2020), CarperAI trlX (Havrilla et al., 2023) and Microsoft DeepSpeed-Chat (Yao et al., 2023). These tools provide an excellent starting point with respect to usability and feature set. However, with NeMo-Aligner we aim to vastly improve performance and scalability to more than a thousand GPUs, especially useful for aligning the largest and most

competent models such as Nemotron 4 340B (Nvidia et al., 2024), Llama 3.1 405B (Dubey et al., 2024) and beyond.

NeMo-Aligner addresses scalability challenges by (I) building upon Megatron-LM (Shoeybi et al., 2020) with 3D (data, tensor, and pipeline)-parallelism training, (II) having a distributed approach to Proximal Policy Optimization (PPO) training in RLHF and (III) integrating PPO inference optimizations based on TensorRT-LLM (NVIDIA, 2023b) during rollout stage. Combined, these optimizations allow users to efficiently train the largest models over a thousand GPUs reducing research iteration time.

NeMo-Aligner optimizes popular alignment techniques including Supervised Finetuning (SFT), PPO-based RLHF (Ouyang et al., 2022), Direct Preference Optimization (Rafailov et al., 2023), SteerLM (Dong et al., 2023) and Self-Play Fine-Tuning (Chen et al., 2024). We briefly outline the background for these techniques in Section 2, followed by an in-depth exploration of training with each of the techniques in Sections 3, 4, 5, and 6. Finally, we demonstrate the extensible design of NeMo-Aligner in Section 7.

## 2   Model Alignment Background

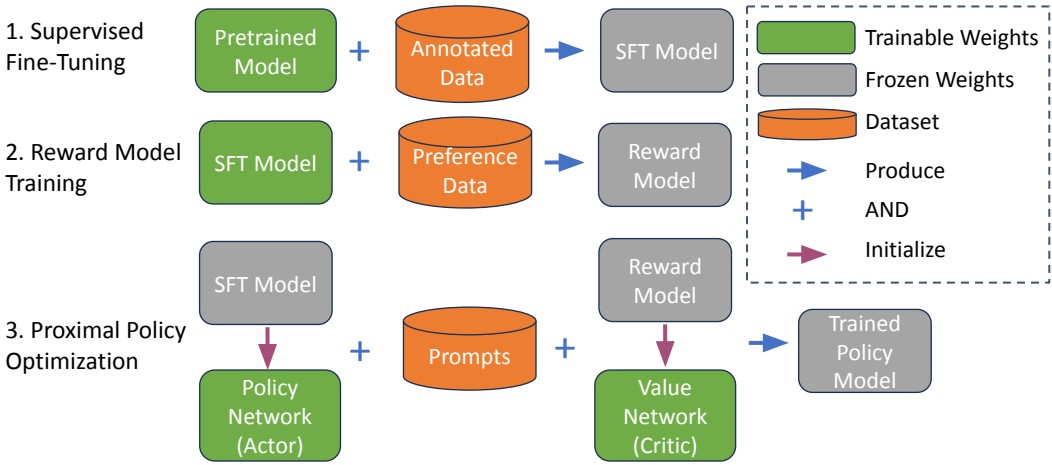

Figure 1: Training Recipe for RLHF based on Ouyang et al. (2022). *Step 1*: Annotated Prompt-Response Data is used to perform Supervised Fine-Tuning on the pre-trained (base) Model. *Step 2*: The resulting SFT model is trained with Preference Data to produce a Reward Model. *Step 3*: The SFT Model is used to initialize the Policy Network, and the Reward Model is used to initialize the Value Network – together with input prompts, all four models are used to train a Policy Model. The SFT model is also used to compute the KL divergence penalty in Step 3 (not illustrated).

### 2.1   Supervised Fine Tuning

Given a pre-trained (also referred to as "base") model, supervised fine-tuning (SFT) updates the base model's parameters on prompts with expected responses, where the expected responses might come from expert human annotations (Köpf et al., 2023) or other language models (Ding et al., 2023). The model is trained to mimic the expected responses given prompts using the token-level cross-entropy loss. SFT is an important prerequisite step in Reinforcement Learning from Human Feedback (Ouyang et al., 2022) and Direct Preference Optimization (Rafailov et al., 2023) because without it, the base model is very unlikely to generate responses which follow user's instructions. This step is also sometimes called *behavior cloning* because the model is expected to mimic responses of a human or another model.

## 2.2 Reinforcement Learning from Human Feedback

Reinforcement Learning from Human Feedback (RLHF) was introduced by Christiano et al. (2017) as a way to avoid manually defined reward functions in Reinforcement Learning. Instead, a reward model is trained from a dataset of human preferences consisting of pairs of "chosen" and "rejected" trajectories. The reward model's loss, derived from the Bradley-Terry model (Bradley & Terry, 1952), tries to maximize the likelihood that $r_{chosen} > r_{rejected}$ (i.e., that the predicted rewards are consistent with human preferences). Once the reward model is trained, it may be used to compute rewards for RL algorithm. Two most common methods used in RLHF are REINFORCE (Williams, 1992) and Proximal Policy Optimization (PPO) (Schulman et al., 2017). In NeMo-Aligner we focus on PPO, specifically as described by Ouyang et al. (2022).

RLHF has been shown to bring significant benefits for model alignment (Ouyang et al., 2022; Bai et al., 2022a; Touvron et al., 2023) with the typical training recipe being as follows, also illustrated in Figure 1:

1. From a pre-trained base model, train an initial SFT model as described in Section 2.1.

2. From the SFT model, train a reward model using a dataset of human preferences made of pairs of "chosen" and "rejected" responses to a set of prompts, following Christiano et al. (2017). Typically, we initialize a linear reward model head on top of the SFT model before training.

3. From the SFT model, train a policy with the online Proximal Policy Optimization algorithm (PPO, Schulman et al., 2017), with rewards provided by the trained reward model. Input prompts may not necessarily be the same as those used for reward model training. A regularization term based on the KL divergence w.r.t. the SFT model helps prevent the policy from straying too far away from its starting point and exploiting the "blind spots" of the reward model (Stiennon et al., 2020; Ouyang et al., 2022). The PPO critic is typically initialized from the reward model.

## 2.3 Direct Preference Optimization

Direct Preference Optimization (Rafailov et al., 2023) is an offline, off-policy algorithm that makes use of preference data to directly train an optimal policy without an explicit reward model. Rather than use a reward model, a reference policy is used to implicitly derive the reward between a chosen and rejected pair via the Bradley-Terry model. This is accomplished via the difference in the log probabilities between the chosen and rejected responses, which is calculated for the optimal and reference policies. This difference is scaled and then transformed by the sigmoid function to derive the loss. The reference policy is frozen during training and represents the policy used to generate the chosen/rejected responses to the given prompts. If the reference policy used to generate the preference data is not available, it can be approximated by supervised fine-tuning on the prompts and preferred responses of the preference data.

## 2.4 SteerLM

SteerLM (Dong et al., 2023) is a model alignment algorithm based on supervised finetuning which avoids use of complex RL methods, similarly to DPO. SteerLM involves three steps. The first step is to train an Attribute Prediction Model that learns to predict the values (between 0 and 4 where higher is more) for various semantic aspects of a response that make responses helpful and safe, such as its correctness and toxicity (Köpf et al., 2023; Wang et al., 2023). Next, the Attribution Prediction Model can be used to annotate the various attributes contributing to helpfulness and safety in a diversity of prompt-response datasets. Finally, these annotated datasets can be used to perform Attribute-Conditioned Supervised Fine-Tuning where the model learns to generate the response conditioned on the prompt as well as the annotated attributes formatted into a string, such as `helpfulness:4,correctness:4,toxicity:0`. This step teaches the model to discriminate

between responses that are more helpful/safe and those that are less, in a fine-grained manner for each semantic aspect. At inference time, the prompt can be appended with the optimal attribute values, as above, to generate the most helpful response.

## 2.5 Self-Play Fine-Tuning

Self-Play Fine-Tuning (SPIN) (Chen et al., 2024) is a self-play based algorithm, where a strong model is developed from a weaker model by playing against previous instances of itself. Starting from an SFT dataset of prompt/response pairs, new responses are generated from previous iterations of the model. Its policy is then improved by discriminating between these self-generated responses and the ground truth human-generated SFT responses. This is accomplished through a preference loss function which is identical to the one used by DPO (Section 2.3). When SPIN training first starts, we use a copy of the initial policy as the reference policy in the DPO loss. The self-play "game" is then played for a number of iterations during which we train the policy as in DPO whilst keeping the reference policy frozen, and at the end of each iteration we update the reference policy's weights with those from the trained policy. During each iteration, we iterate over our SFT training dataset and use the reference policy to generate responses for each prompt, building a preference tuple between the ground truth SFT human "chosen" response and the generated "rejected" response. Once we have these preference tuples for the entire epoch, we update the model weights via the DPO loss function from these tuples of "(chosen, rejected)" preference pairs. The model thus implicitly learns to prefer the ground truth SFT responses to those generated by the previous iteration of itself, which forms the self-play mechanism.

## 3 RLHF (PPO) Training

NeMo-Aligner is designed to support numerous alignment techniques efficiently at extremely large scales. It does so by building upon Megatron-LM (Shoeybi et al., 2020) and NeMo (Kuchaiev et al., 2019) to include features such as optimized kernels from Transformer Engine (NVIDIA, 2022), distributed fused adam optimizer and 3D parallelism support. NeMo-Aligner supports the entire RLHF pipeline as introduced by Ouyang et al. (2022) and described in Section 2.2. The training pipeline is separated into three distinct stages as illustrated in Figure 1: Supervised Fine-Tuning, Reward Model Training, and Proximal Policy Optimization. The challenges with the pipeline efficiency come primarily from the Proximal Policy Optimization stage, and this section describes our approach to tackling these challenges, as summarized in Figure 2.

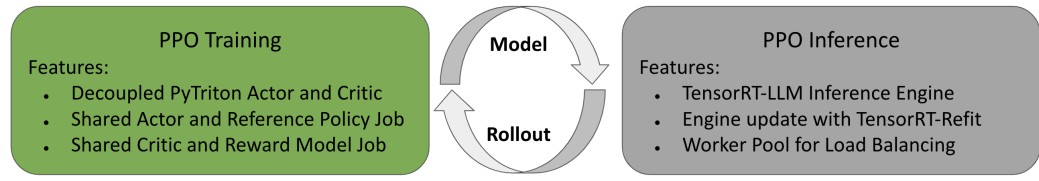

Figure 2: Optimizations for RLHF training. Optimizations for PPO training and inference are detailed in Sections 3.1 and 3.2 respectively.

## 3.1 Distributed Approach to PPO training

The PPO stage requires running training and/or inference on four different models, as illustrated in Figure 3:

1. PPO Actor (training and inference, initialized from SFT model): The model we want to fine tune with PPO.
2. Reference Policy (inference only, set to the SFT model): The model to compute the KL penalty against.

3. PPO Critic (training and inference, initialized from the reward model): Used in PPO to compute value estimates.

4. Reward Model (inference only) : Provides RL rewards on generated rollout data.

All of these models can be extremely large (e.g. Llama 3.1 405B), so NeMo-Aligner takes a distributed approach to PPO training. We allow users to setup PyTriton (NVIDIA, 2022) servers and clients to communicate across the different models during PPO. These PyTriton servers make it possible to run the models on different compute clusters, removing the requirement of having both the critic and actor on the same compute allocation. Naively, four different servers (i.e. one for each model) would be launched. However, we note that the reference policy and PPO actor are the same model but with different weights. Therefore, we combine them into one job and offload the reference policy's weights to CPU, swapping them with the actor's weights for the reference policy inference step. We deploy the same strategy for the reward model and critic. All communications are done asynchronously, permitting pipelined critic inference/training with policy inference/training.

We scale compute allocation sizes such that the [reward model inference + critic inference] ≈ [actor sampling + reference policy inference] and [critic train] ≤ [actor train + actor inference initialization]. This ensures that the pipeline can use available compute capacity most efficiently.

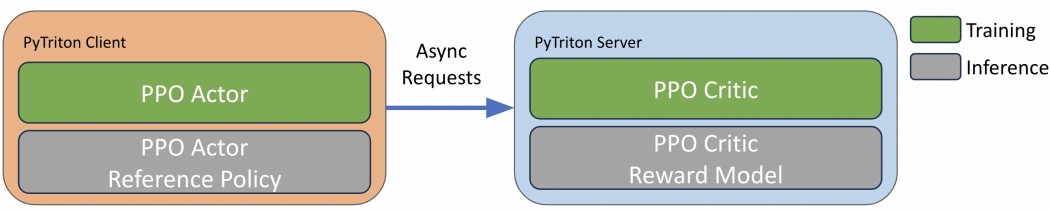

Figure 3: NeMo-Aligner PPO System Architecture. The PPO Actor is a PyTriton (NVIDIA, 2022) client that sends async requests to the server (PPO critic and reward model) to obtain the rewards and values of generated rollouts, and to send the training data for the critic.

## 3.2 Optimizations for PPO rollout

Response generation during the rollout step dominates the end to end time of PPO training. The generation stage of the actor is composed of multiple forward passes, with one token generated per forward pass. Therefore, generation stage kernels are generally launch latency and memory bandwidth bound, meaning that directly reusing the compute optimized forward pass implementation of the training stage results in very poor performance.

To address these bottlenecks, we implement the generation stage using TensorRT-LLM (NVIDIA, 2023b), a high-performance LLM deployment framework. TensorRT-LLM integrates inference optimized kernels and automatic kernel fusion into a TensorRT based runtime to achieve better performance. At the start of RLHF, the model is passed to TensorRT-LLM which compiles the model into a TensorRT engine; TensorRT-LLM loads the engine into its runtime and performs generation. The engine holds a copy of the model weights along with the runtime KV-cache and activations. Due to the cost of serializing the engine, we keep the engine in memory during training. As a result, we reduce the peak memory pressure by recomputing training stage activations in the backward pass. Furthermore, because generation has lower memory requirements than training, we reshard the model to only use tensor parallelism during inference if memory allows, removing overhead from inter-node communication when otherwise running with pipeline parallelism.

On subsequent training steps, the engine must be synced with updated parameter weights from the training stage. The engine is updated in-place using the TensorRT Refitter (NVIDIA, 2023c). We avoid recompiling the engine which would incur a large overhead as generation cannot begin until the weights are updated.

For large problem sizes, there may be discrepancies of generation time between the fastest and the slowest data parallel worker during generation due to the differences in response lengths. To mitigate this, we allow users to setup a worker pool to dynamically load balance among data parallel workers to give workers with shorter generations correspondingly more work.

### 3.3 Model Training Details and Quality

As a demonstration of practical large-scale RLHF training with NeMo-Aligner, we train Llama3 (Meta AI, 2024) 70B model using PPO as prescribed in Wang et al. (2024). The PPO model is trained with a rollout global batch size of 128, training global batch size of 128, constant learning rate of 1e-7, KL penalty of 0.003, and using HelpSteer2 as a prompt source.

Following Wang et al. (2024); Meng et al. (2024), we use MT-Bench (Zheng et al., 2023a) with GPT-4-Turbo judge to evaluate the performance of the trained RLHF Model. The resulting model achieves a performance of 8.13 on MT-Bench, which is an improvement upon the strong starting SFT checkpoint with MT-Bench of 7.96. The Reward Model and RLHF-trained model are publicly released at `https://huggingface.co/nvidia/Llama3-70B-SteerLM-RM` and `https://huggingface.co/nvidia/Llama3-70B-PPO-Chat` respectively.

### 3.4 Scalability

| No. of compute nodes (Actor + Critic) | 8 + 4 | 16 + 8 |
|---|---|---|
| *Time per step in seconds (std.)* ↓ | | |
| Overall | 53.7 (2.223) | 29.8 (1.459) |
|   Train | 8.5 (0.356) | 4.8 (0.118) |
|   Rollout | 45.2 (1.957) | 25.0 (1.41) |
|     - Response generation | 35.8 (1.658) | 17.5 (1.224) |
|     - Log-probs calculation | 4.0 (0.573) | 2.8 (0.275) |
|     - TensorRT Refit | 3.1 (0.182) | 2.3 (0.334) |
|     - Critic wait | 0.01 (0.001) | 0.01 (0.001) |
| *Relative speed up (vs. 8 + 4 node setup)* ↑ | | |
| Overall | 1x | 1.80x |
|   Train | 1x | 1.77x |
|   Rollout | 1x | 1.81x |
|     - Response generation | 1x | 2.05x |
|     - Log-probs calculation | 1x | 1.43x |

Table 1: Effects of scaling training across different number of compute nodes for Llama 3 70B actor and Llama 3 70B critic on rollout global batch size of 128 and BF16 precision following Section 3.3. Nodes are 8*H100-80GB-SXM connected with intra-node NVLink (NVIDIA, 2023a) and inter-node Infiniband (NVIDIA, 2024) interconnects. Time per step calculated based on mean of 5 steps after the first step, as the first step incurs substantial time for TRT-LLM Engine Building. Further training configuration details are in Table 3.

To demonstrate the scaling efficiency of NeMo-Aligner, we repeat identical training setups from Section 3.3, with 8 actor nodes + 4 critic nodes and 16 actor nodes + 8 critic nodes. As shown in Table 1, overall time per step reduces correspondingly, achieving a 1.80x speed up between 8+4 nodes and 16+8 nodes. The speed up in overall time per step is contributed by speed ups in both the Train and Rollout stages, demonstrating the effective optimization that NeMo-Aligner has done for both stages.

The scaling of Train stage is sublinear due to number of micro-batches per data parallel rank decreasing as node count increases. Because all pipeline stages must complete before the optimizer is called in pipeline parallel models, we incur an overhead to fill and drain the pipeline that is independent of the number of micro-batches (Shoeybi et al., 2020). Therefore, decreasing the number of micro-batches per data parallel rank increases the proportion of

the train step spent in filling and draining the pipeline, where GPU utilization is poor. A similar issue is apparent during the log prob calculation phase with scaling of 1.43x.

| No. of compute nodes (Actor + Critic) | 16 + 8 | 32 + 16 | 64 + 32 |
|---|---|---|---|
| *Time per step in seconds (std.)* ↓ | | | |
| Overall | 190.4 (15.392) | 106.8 (6.842) | 56.9 (7.596) |
| Train | 38.8 (4.674) | 22.2 (2.408) | 14.1 (1.439) |
| Rollout | 151.5 (13.172) | 84.6 (5.596) | 42.7 (6.36) |
| - Response generation | 131.3 (12.551) | 71.7 (5.093) | 29.9 (6.456) |
| - Log-probs calculation | 15.5 (3.055) | 8.5 (1.753) | 6.2 (1.177) |
| - TensorRT Refit | 2.2 (0.408) | 2.2 (0.014) | 3.5 (0.063) |
| - Critic wait | 0.02 (0) | 0.03 (0.001) | 1.7 (2.045) |
| *Relative speed up (vs. 16 + 8 node setup)* ↑ | | | |
| Overall | 1x | 1.78x | 3.35x |
| Train | 1x | 1.75x | 2.75x |
| Rollout | 1x | 1.79x | 3.55x |
| - Response generation | 1x | 1.83x | 4.39x |
| - Log-probs calculation | 1x | 1.82x | 2.50x |

Table 2: Effects of scaling training for Llama 3 70B actor and Llama 3 70B critic on rollout global batch size of 1024 and BF16 precision. Nodes are 8*H100-80GB-SXM connected with intra-node NVLink (NVIDIA, 2023a) and inter-node Infiniband (NVIDIA, 2024) interconnects. Time per step calculated based on mean of 5 steps after the first step, as the first step incurs substantial additional time for TRT-LLM Engine Building. Our distributed optimizer feature enables NeMo-Aligner to support double the rollout microbatch size from 8 to 16 in the 64 + 32 node configuration. This speeds up the response generation time significantly. Further configuration details are in Table 3.

Generation time scales well with the number of nodes, achieving a near-linear speedup of 2.05x. This is because scaling up the number of actor nodes proportionally increases the number of data parallel workers for each step, which can evenly share the intense work of generation. However, the time spent fitting weights into the TensorRT engine remains relatively constant and therefore does not scale well with node count. Finally, async communications between the Actor and the Critic models result in the additional time taken to wait for the Critic model to be inconsequential (0.01 seconds), suggesting the effectiveness of having async non-blocking calls between actor and critic models in the PPO pipeline.

| Model | No. of compute nodes | Tensor Parallel | Pipeline Parallel | Data Parallel | Rollout Batch Size Micro | Global |
|---|---|---|---|---|---|---|
| 70B + 70B | 8 + 4 | 8 + 8 | 8 + 4 | 1 + 1 | 8 | 128 |
| 70B + 70B | 16 + 8 | 8 + 8 | 8 + 4 | 2 + 2 | 8 | 128 / 1024 |
| 70B + 70B | 32 + 16 | 8 + 8 | 8 + 4 | 4 + 4 | 8 | 1024 |
| 70B + 70B | 64 + 32 | 8 + 8 | 8 + 4 | 8 + 8 | 16 | 1024 |
| 405B + 405B | 84 + 42 | 8 + 8 | 21 + 21 | 4 + 2 | 16 | 128 |

Table 3: Parallelism settings for scaling experiments on Llama3 models. The node counts and parallelism configurations are denoted as actor + critic.

System scalability also needs to be considered under the context of the problem requirements. The training setup in Section 3.3 has a 70B Llama 3 Actor, 70B Llama 3 Critic as well as a rollout global batch size of 128. Such a setup limits the effective demonstration of our system scaling beyond 16 + 8 nodes as there is not enough work to be meaningfully shared across more data parallel workers. Therefore, we modify the setup slightly to use a rollout global batch size of 1024 in Table 2 in order to measure the system performance when the requirements are higher. Table 2 shows that the increased requirements of the training job allows it to meaningfully scale to 64 + 32 nodes (with 768 H100 GPUs total) for various stages within PPO.

### 3.5 What contributes to system performance?

|  | Time per step in seconds (std.) ↓ | Time relative to Optimal RLHF setup ↓ |
|---|---|---|
| Optimal RLHF Setup | 53.7 (2.223) | 1x |
| - TensorRT-LLM Integration (*i.e.* using NeMo Generate) | 372.2 (37.433) | 6.93x |
| - Reshard | 207.6 (8.940) | 3.87x |
| - TensorRT Refit | 169.0 (5.206) | 3.15x |
| - Async Requests | 82.8 (7.826) | 1.54x |

Table 4: Ablation studies on training Llama 3 70B actor and critic on rollout global batch size of 128 with 8 Actor nodes and 4 critic nodes. Nodes are 8*H100-80GB-SXM connected with intra-node NVLink (NVIDIA, 2023a) and inter-node Infiniband (NVIDIA, 2024) interconnects. Time per step is calculated based on mean of 5 steps after the first step, as the first step incurs substantial additional time for TRT-LLM Engine Building.

To better understand the importance of each aspect of NeMo-Aligner's PPO system design, we conduct ablation studies by removing one aspect at a time and measuring the overall time per step as shown in Table 4. We find that TensorRT-LLM Integration is the most critical component for high system performance, without which PPO will take nearly seven times as long for each step. This is followed by resharding our model to use tensor parallelism only during inference (3.87x), using TensorRT Refit to avoid TensorRT-LLM engine recompiling (3.15x), the use of async requests between actor and critic models (1.54x). We did not observe meaningful speedup in using the worker pool to balance the work between data parallelism ranks because the problem size is small (with rollout global batch size of 128) and therefore the worker imbalance is less than the overhead we incur doing the balancing itself. Nevertheless, we expect this feature to help for larger problem sizes.

### 3.6 Training the largest open source LLM

| No. of compute nodes (Actor + Critic) | 84 + 42 |
|---|---|
| *Time per step in seconds (std.)* ↓ |  |
| Overall | 164.60 (3.434) |
|     Train | 5.60 (0.532) |
|     Rollout | 158.90 (2.969) |
|       - Response generation | 140.10 (1.067) |
|       - Log-probs calculation | 17.20 (2.352) |
|       - TensorRT Refit | 0.70 (0.108) |
|       - Critic wait | 0.30 (0.454) |

Table 5: NeMo Aligner supports the largest open source LLM as of July 2024 (Llama 3.1 405B) as both the actor and critic. We run the model on rollout global batch size of 128 and BF16 precision. Nodes are 8*H100-80GB-SXM connected with intra-node NVLink (NVIDIA, 2023a) and inter-node Infiniband (NVIDIA, 2024) interconnects. Time per step is calculated based on mean of 5 steps after the first step, as the first step incurs substantial additional time for TRT-LLM Engine Building. Further configuration details are in Table 3.

NeMo-Aligner supports alignment of the largest open source LLMs as of July 2024, such as Nemotron 4 340B (Nvidia et al., 2024) and Llama 3.1 405B (Dubey et al., 2024). In Table 5, we perform PPO on Llama 3.1 405B using 1008 H100 GPUs, using a configuration based on Sec. 3.3. We use Llama 3.1 405B Instruct as the actor and we train a Reward Model on top of Llama 3.1 405B Instruct using the same data and hyper-parameters as Nemotron 4 340B Reward (Wang et al., 2024). Compared to the 70B model, the 405B model is substantially slower to train during PPO, mainly bottle-necked by the slow response generation stage.

This is because the 405B model generation cannot fit into a single node and requires pipeline-parallel generation which we plan to optimize further in future work. Additionally, in the first few steps of PPO, the 405B model generates longer responses (mean of 916 tokens) than the 70B model with a mean of 351 tokens.

# 4 DPO Training

We follow the Zephyr-7B-Beta (Tunstall et al., 2023) training recipe, a model trained with SFT and DPO. Briefly, SFT was first performed on Mistral-7B (Jiang et al., 2023) using the Ultrachat dataset (Ding et al., 2023). Model was then further trained with DPO using the Ultrafeedback dataset (Cui et al., 2023). For SFT, we used a constant learning rate of $2e-5$, global batch size of 512, and trained the model for 3 epochs. For DPO training, we used KL regularization coefficient of $3e-4$, global batch size of 512 and a cosine learning rate schedule with peak LR of $1e-7$, minimum LR of $1e-8$, 50 warmup steps, and max. 300 steps. We obtain slighter better MT-Bench scores than those reported by Tunstall et al. (2023) for both the final model (7.60 vs 7.34) and the SFT-only initial model (6.77 vs 6.64).

# 5 SteerLM Training with LoRA

Low Rank Adaptation (Hu et al., 2021) enables fine-tuning large language models in a more efficient and cost-effective manner. Supported for various alignment techniques within NeMo-Aligner, LoRA is applied to SteerLM training following the training recipe by Wang et al. (2023) using the Llama 2 70B model as well as the HelpSteer (Wang et al., 2023) and Open Assistant datasets (Köpf et al., 2023). Specifically, we applied LoRA to all attention layers, with a rank of 32. We used global batch size of 128, constant learning rate of $1e-5$ after 10 warmup steps with the AdamW optimizer, and trained for 3 epochs. As shown in Table 6, applying LoRA to SteerLM training with BF16 can reduce the minimum number of 80GB GPUs required from 32 to 8. With the same number of GPUs, LoRA achieves a $5\times$ speedup compared to full-parameter fine-tuning, while maintaining comparable model performance: MT-Bench 7.43 vs. 7.54, which is within noise level for this benchmark (Jiang et al., 2023).

|  | Full-Param | LoRA |
|---|---|---|
| # trainable params | 70B | 89M |
| min # 80GB GPUs required | 32 | 8 |
| Relative speed (sample/GPU/s) | $1\times$ | $5\times$ |
| MT-Bench | 7.54 | 7.43 |

Table 6: Comparison of Full-Parameter and LoRA SteerLM following training recipe by Wang et al. (2023).

As we increase the number of GPUs used for LoRA training, the relative throughput (measured in samples per second) improves almost proportionally, as shown in Figure 4. This shows that NeMo-Aligner can effectively distribute and parallelize the workload across a large number of GPUs with minimal overhead and diminishing returns.

# 6 SPIN Training

We recreate the Zephyr-7B-Beta (Tunstall et al., 2023) SFT model via SPIN instead of SFT as formulated by Chen et al. (2024). We start with the Mistral-7B base model (Jiang et al., 2023) and perform SPIN training following Chen et al. (2024). However, we make a few departures from their methodology, in that we do not inject generations from the previous iteration into the current iteration (which would double the dataset size every epoch), and we only train for a single iteration, with 1 epoch per iteration. Additionally, we use a random subset of only 50k samples from Ultrachat200k (Ding et al., 2023) rather than the entire

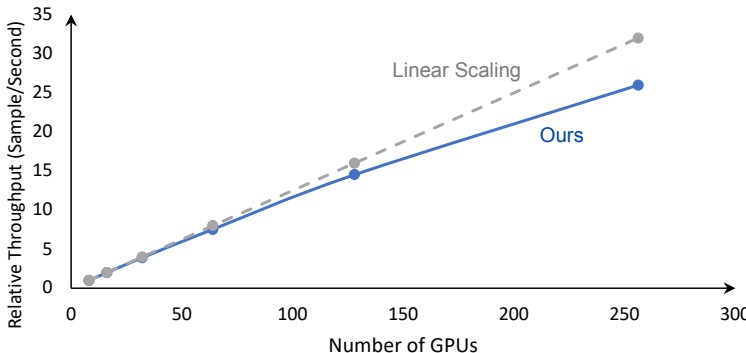

Figure 4: Relative throughput of LoRA applied to SteerLM training as the number of GPUs increases.

dataset, and use AdamW instead of RMSProp. Our learning rate is $5e - 7$ with 400 total steps, 40 warmup steps, and this LR is then decayed to $1e - 7$ for the last 100 steps using cosine annealing. Global batch size is 64, weight decay is 0.0, and the KL regularization coefficient is 0.1, as per Chen et al. (2024). Using this approach, we achieve an MT-Bench score of 7.04 which exceeds the 6.64 of Zephyr-7B-Beta using SFT (Tunstall et al., 2023), as well as the 6.78 of the 3-iteration SPIN model (Chen et al., 2024).

# 7 Framework Extensibility

We design NeMo-Aligner with extensibility in mind, allowing users to easily modify algorithms in spite of the complexities of distributed training. We do so using the trainer abstraction, which encourages re-use of existing trainer methods across various steps and approaches. The extensibility of NeMo-Aligner allows variants of DPO to be integrated with minimal code changes, including the Identity Preference Optimization (Azar et al., 2023), the Conservative DPO (Mitchell, 2023), and the Kahneman-Tversky Optimization (Ethayarajh et al., 2023). Furthermore, other model alignment techniques such as Constitutional AI (Bai et al., 2022b), Rejection Sampling (Touvron et al., 2023), and Self-Rewarding Language Models (Yuan et al., 2024) are also being incorporated into NeMo-Aligner, facilitated by the framework design.

# 8 Conclusion

Modern model alignment techniques, especially those based on Reinforcement Learning, pose complex optimization challenges with respect to system implementation. We create and open-source NeMo-Aligner to allow AI researchers and practitioners to efficiently experiment with LLM alignment by utilizing all available compute in a scalable way. Our framework consistently scales well when training large models with more compute. As this is our initial release, we expect this scaling to only improve with future versions. Additionally, we support SFT, PPO, DPO, SteerLM in a parameter-efficient manner using LoRA for compute-limited settings. As an Apache 2.0 licensed open-source codebase, NeMo-Aligner can make alignment research more efficient and accessible.

# Acknowledgements

We would like to thank many teams at NVIDIA who contributed towards enabling NeMo-Aligner, especially the NeMo, TRT-LLM and TensorRT teams.

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
