# OpenReview forum: "NeMo-Aligner: Scalable Toolkit for Efficient Model Alignment"
_colmweb.org/COLM/2024/Conference — COLM_

### Official Review · Reviewer_X9GJ · 2024-05-10

**Rating:** 6
**Confidence:** 3
**Ethics Flag:** 1

**Summary:**

This paper introduces STEMA, a scalable toolkit for efficiently aligning large language models (LLMs) using techniques like Reinforcement Learning from Human Feedback (RLHF), Direct Preference Optimization (DPO), SteerLM, and Self-Play Fine-Tuning (SPIN). STEMA builds upon the Megatron-LM framework and incorporates optimizations to enable efficient training across hundreds of GPUs. Key innovations include a distributed approach to Proximal Policy Optimization (PPO) training in RLHF, integration with TensorRT-LLM for faster generation during rollouts, and support for parameter-efficient fine-tuning using methods like LoRA. The authors demonstrate STEMA's scalability and performance gains through experiments aligning Llama 2 models up to 70B parameters.

**Reasons To Accept:**

- Achieves impressive scalability, enabling alignment of models with over 70B parameters across numerous GPUs.
- Incorporates optimizations tailored to the unique compute challenges of alignment techniques like RLHF.
- Supports a range of state-of-the-art alignment methods in one extensible framework.

**Reasons To Reject:**

- While supporting multiple alignment techniques is a strength, the paper could benefit from a more in-depth comparison of the pros and cons of the different frameworks, such as TRL. Especially the pain points in the previous framework should be discussed more to understand how the suggested framework improves or mitigates the points.
- The model training details were too simplified. For example, the training dataset mixture might be a more important factor for the final performance. Thus, the authors should present the details of the dataset mixture. Furthermore, in the same sense, the SFT model should also be compared to the final PPO model and LLama-70B-Chat.

---

> ### Author Rebuttal · Authors · 2024-05-30
>
> Thank you for your detailed appraisal of our paper!
>
> 1. We agree that it is important to make comparisons with other existing alignment toolkits. However, it is difficult to make such comparisons fairly given public information. For instance, the papers for CarperAI trlX (Havrilla et al., 2023) and Microsoft DeepSpeed Chat (Yao et al., 2023) report demonstrates good scaling up to 64 GPUs (8 nodes of 8 GPUs each). Huggingface TRL does not have a paper but likely does not scale as well as trlX since trlX was a fork off from TRL to support larger models. On the other hand, we demonstrate that STEMA can scale up to 768 GPUs (64 + 32 nodes of 8 GPUs). However, it’s not clear whether they do not scale to even more GPUs or simply did not have sufficient resources to try scaling. Therefore, we were cautious not to make such definitive claims. Nonetheless, it is plausible that these frameworks support about an order of magnitude fewer GPUs than STEMA. The implication of supporting more GPUs is that while STEMA allow full-parameter optimization of Llama 2 70B within a short period of time (e.g. a few hours), other frameworks have to resort to various compromises (such as LoRA, shared parameters between actor and critic, freezing a proportion of weights) in order to train such a model and still taking substantial time.
>
> 2. We agree with your concerns and would like to provide more info about our SFT data blend. Specifically, our public data component includes 128,000 samples from Open Assistant, CodeContests, FLAN, PRM800K, GSM8K, Sci-Bench, tigerbot-kaggle-leetcodesolutions-en-2k, OpenbookQA, and ARB. Additionally, we incorporated around 5000 proprietary data samples covering extraction, writing, classification, and question-answering tasks. We have open-sourced the public data mix on Huggingface but avoided including the link in our submission or author response to maintain anonymity.  We also appreciate your suggestion to compare our SFT model to the final PPO model and LLama-70B-Chat. Our SFT model gets an MT-Bench score of 6.82, which is slightly worse than LLama-70B-Chat(6.86). The final PPO model gets 7.59, which shows the effectiveness of our PPO implementation. We will make sure to include these comparisons in the revised version of our paper, providing a more comprehensive evaluation of our model's performance.
>
> We hope we have sufficiently addressed your concerns and would like for you to reconsider your rating for our paper if you find so too.

---

> > ### Comment · Reviewer_X9GJ · 2024-06-06
> >
> > Thank you for the responses. I will keep my score.

---

### Official Review · Reviewer_VzBx · 2024-05-11

**Rating:** 7
**Confidence:** 3
**Ethics Flag:** 1

**Summary:**

This paper builds an effective tool (STEMA) for aligning large-scale language models. By utilizing technologies such as Megatron LM and TensorRT LM, they have built implementations of popular RLHF methods, including PPO, DPO, SteerLM, and SPIN. The preliminary experimental results on MT-Bench show that it can make the alignment of LLMs and achieve better results through this framework. The paper also elaborates on analyzing the operational details of the entire framework and emphasizes its scalability.

**Questions To Authors:**

See the above in the Reasons To Reject.

**Reasons To Accept:**

The construction of infrastructure and engineering frameworks for large language models is crucial as it can promote the progress of the entire research community.

The author demonstrated the effectiveness of the framework through experiments and demonstrated its scalability.

**Reasons To Reject:**

Lack of comparison with existing technology. The author mentioned existing frameworks such as TRL, trlX, and DeepSpeedChat in the Introduction and claimed that “However, with STEMA we aim to vastly improve performance and scalability of PPO and other methods especially when it comes to the largest and most competent models such as Llama 2 70B (Touvron et al., 2023) and beyond.” Do these exsiting works cannot optimize the Llama 2 70B? Or do they have some shortcomings when optimizing? More time-consuming, higher-cost, or higher requirements for equipment? If there are some comparisons between STEMA and other frameworks to show the strongness of STEMA, I will accept it clearly.

Another reason why I did not provide a clear acceptance is that the paper did not propose technological innovation and relied on the combination of existing technologies. They have made engineering contributions but lack deeper scientific contributions.

---

> ### Author Rebuttal · Authors · 2024-05-30
>
> Thank you for your meticulous review of our paper.
>
> 1. We agree that it is important to make comparisons with other existing alignment toolkits. However, it is difficult to make such comparisons fairly given public information. For instance, the papers for CarperAI trlX (Havrilla et al., 2023) and Microsoft DeepSpeed Chat (Yao et al., 2023) demonstrate good scaling up to 64 GPUs (8 nodes of 8 GPUs each). For context, Huggingface TRL does not have a paper but likely does not scale as well as trlX since trlX was a fork off from TRL to support larger models. On the other hand, we demonstrate that STEMA can scale up to 768 GPUs (64 + 32 nodes of 8 GPUs). However, it’s not clear whether other libraries do not scale to even more GPUs or simply did not have sufficient resources to try scaling. Therefore, we were cautious not to make such definitive claims. Nonetheless, it is plausible that these frameworks support about an order of magnitude fewer GPUs. The implication of supporting more GPUs is that while STEMA allow full-parameter optimization of Llama 2 70B within a short period of time (e.g. a few hours), other frameworks have to resort to various compromises (such as LoRA, shared parameters between actor and critic, freezing a proportion of weights) in order to train such a model and still taking substantial time.
>
> 2. We believe our framework proposes several core innovations that go beyond combining existing technology.  For instance, prior to our framework, many have found the rollout/generation stage within PPO to be slow. However, separating out the training engine and the inference engine within PPO was not practical until we proposed and implemented runtime-update of inference engine weights using TensorRT Refit. This involved connecting to low-level TensorRT APIs in a bespoke manner to avoid engine compilation and loading overheads, which naive integrations would incur since training and inference engines independently allocate memory. Similarly, enabling interleaved, asynchronous communication between actor and critic nodes did not work OOTB with PyTriton and instead required us to modify the underlying PyTriton client to facilitate low-latency, high-throughput, inter-node calls between the actor and critic. Considering the audience of CoLM (mostly NLP researchers), we previously left out further system-level orchestration details for clarity but we would take care to include it in a later version.

---

> > ### Comment · Reviewer_VzBx · 2024-06-04
> > **Thank you for your reply**
> >
> > Thank you for your reply. I think my score is good enough, and I will keep my score.

---

### Official Review · Reviewer_mnwD · 2024-05-11

**Rating:** 7
**Confidence:** 2
**Ethics Flag:** 1

**Summary:**

This work presents an ope-sourced toolkit that implements several major paradigms of model alignment such as: Reinforcement Learning from Human Feed-back, Direct Preference Optimization, SteerLM, and Self-Play Fine-Tuning. The training pipeline is divided in three steps: supervised fine-tuning, reward model training and proximal policy optimization. The toolkit focus on improving performance, the most critical components being TensorRT-LLM integration to mitigate the bottleneck of response generation during the rollout step; using TensorRT Refit to avoid TensorRT-LLM engine recompiling; using async requests between actor and critic models; and the load-balancing of data parallel workers during generation using a worker pool.

**Reasons To Accept:**

A useful tool that can run several alignment techniques in a parameter efficient tuning setting and is designed for extensibility.

**Reasons To Reject:**

Instead of being part of section 3, a new section should be created for the evaluation, extending the level of  detail of the experimental setup (e.g., the only description of the training dataset is "using a mixture of public and proprietary data").

---

> ### Author Rebuttal · Authors · 2024-05-30
>
> Thank you for taking the time to review our paper and providing feedback. We are pleased that you found our tool to be useful and extensible.
>
> Regarding your suggestion to create a new section for the evaluation, we understand your point and acknowledge that the experimental setup could be further elaborated. We would like to provide more details on our SFT training dataset, which consists of a mixture of public and proprietary data. Specifically, our public data component with 128,000 samples includes OASST[1], CodeContests[2], FLAN[3], PRM800K[4], GSM8K[5], Sci-Bench[6], tigerbot-kaggle-leetcodesolutions-en-2k[7], OpenbookQA[8], and ARB[9]. Additionally, we incorporated approximately 5000 proprietary data samples covering extraction, writing, classification, and question-answering tasks. We have open-sourced the public data mix on Huggingface but avoided including the link in our submission or author response to maintain anonymity. We will ensure that additional information and huggingface link is provided in the revised version of our paper.
>
>
> [1] https://huggingface.co/datasets/OpenAssistant/oasst1
> [2] https://github.com/google-deepmind/code_contests
> [3] https://github.com/google-research/FLAN
> [4] https://github.com/openai/prm800k
> [5] https://github.com/openai/grade-school-math
> [6] https://github.com/mandyyyyii/scibench
> [7] https://huggingface.co/datasets/TigerResearch/tigerbot-kaggle-leetcodesolutions-en-2k
> [8] https://github.com/allenai/OpenBookQA
> [9] https://github.com/TheDuckAI/arb

---

> > ### Comment · Reviewer_mnwD · 2024-06-04
> >
> > Thanks for your clarifications.

---

### Official Review · Reviewer_i2fQ · 2024-05-16

**Rating:** 6
**Confidence:** 4
**Ethics Flag:** 1

**Summary:**

The authors propose a toolbox that implements alignment techniques such as Reinforcement Learning from Human Feedback (RLHF), Direct Preference Optimization (DPO), SteerLM, and Self-Play Fine-Tuning (SPIN). The approach builds upon existing frameworks such as HuggingFace TRL, CarperAI trlX, and Microsoft DeepSpeed-Chat. The toolbox builds upon Megatron-LM with 3D (data, tensor, and pipeline)-parallelism training, introduces a distributed approach to Proximal Policy Optimization (PPO) training in RLHF and integrates PPO inference optimizations based on TensorRT-LLM.

**Reasons To Accept:**

1. Implements and integrates various alignment techniques commonly used by the community, in one toolkit.
2. Makes alignment techniques (like RLHF) more scalable by incorporating a distributed training approach.
3. Proposes an extensible framework that can allow the addition of new alignment methods (like variants of DPO) with minimal code changes.
4. Includes support for Parameter-efficient training methods for algorithms that are a part of the toolbox like  SFT, PPO, DPO, and SteerLM, making it more accessible to the community.

**Reasons To Reject:**

1. The toolbox doesn’t bring significant novelty in terms of contribution and combines already existing implementations and optimizations proposed by the community. I am not sure how adaptable this will be given the massive user base and support community of existing framrworks.
2. The paper doesn’t present a clear motivation for the creation of a framework when well-supported libraries for each alignment technique exist.
3. The paper doesn’t discuss the evaluation and testing components in detail, which can be a significant factor for adoption for the community.
4. Authors don't justify their model choices for benchmarking their suite and don't include a comparison of their performance with existing alignment frameworks.

---

> ### Author Rebuttal · Authors · 2024-05-30
>
> Thank you for your thorough review.
>
> 1 and 2. We agree that it is important to make comparisons with existing alignment toolkits. However, it is difficult to make such comparisons fairly given public information. For instance, the papers for CarperAI trlX (Havrilla et al., 2023) and Microsoft Deepspeed Chat (Yao et al., 2023) demonstrate good scaling up to 64 GPUs (8 nodes of 8 GPUs each). Huggingface TRL does not have a paper but likely does not scale as well as trlX since trlX was a fork off from TRL to support larger models. On the other hand, we demonstrate that STEMA can scale up to 768 GPUs (64 + 32 nodes of 8 GPUs). However, it’s not clear whether other libraries cannot scale to more GPUs or simply did not have sufficient resources to try. Therefore, we were cautious not to make such definitive claims. Nonetheless, it is plausible that these frameworks support about an order of magnitude fewer GPUs, given that they have to resort to various compromises (such as LoRA, shared parameters between actor and critic, freezing a proportion of weights) in order to train the largest models. This situation is complicated by many having extensive private forks off trlX or DeepSpeed in order to support non-native features (e.g. more GPUs or larger models), but given commercial reasons, not disclosing information about these forks. While existing frameworks can still serve parts of the community well, STEMA provides an OOTB solution for aligning the largest, most capable models (>=70B), which existing frameworks may fall short on.
>
> 3. Our paper aims to provide an overview of a scalable alignment toolkit (as a submission to the Engineering for LLM track). As such, we avoided going into extensive evaluation details that are typically found in an empirical research paper. Nonetheless, we have added more details in our response to reviewer mnwD (due to word limit here) - please let us know if you would like more details.
>
> 4. We chose Llama 2 70B, which at the time of submission (Mar 2024), was the most popular large model that many were interested to align and hence focused our evaluation on this model. We agree that reporting a comparison of performance with other toolkits would be ideal to demonstrate the value add. However, this was not feasible due to company policy outside of the engineering and research teams’ control.
>
> We hope we have sufficiently addressed your concerns and would like for you to reconsider your rating for our paper if you find so too.

---

> > ### Comment · Reviewer_i2fQ · 2024-06-06
> >
> > Thank you for your response. I have increased my score given your replies which I believe are justified. I request you also to make these clarifications in the next version.

---

### Decision · Program_Chairs · 2024-07-10

**Decision:**

Accept

**Comment:**

This paper proposes a framework for scaling up several post-training algorithms, including PPO, DPO, SteerLM, and self-play fine-tuning. The paper shows that the framework supports up to several hundred GPUs, with near-linear speedup in the number of GPUs. The reviewers generally seem to feel that the framework will provide useful and extensible infrastructure. A potential weakness was lack of clear comparison with prior work; the authors argue this was not possible because prior work did not attempt to evaluate at this scale. One response (https://openreview.net/forum?id=yK2eGE8QVW&noteId=rZrkVguEso) adds important details about the SFT stage, which should be included in the next version.

[comments from the PCs] Please follow up on the AC recommendations, especially with details about the SFT stage.